# Leveraging tuberculosis programs for future pandemic preparedness: A retrospective look on COVID-19

Whitney Bowen[1], Ho Tri[1], Sebastian Romero[1], Roaa Shaheen[1], Victoria Kipngetich[1], Nick McGowan[1], Sungho Moon[1], Esha Bhattacharya[1], Robert Hecht[2], Shan Soe-Lin[1], Chris Collins[3]*

**1** Jackson School of Global Affairs, Yale University, New Haven, Connecticut, United States of America, **2** Yale Institute for Global Health, Yale School of Public Health, New Haven, Connecticut, United States of America, **3** Friends of the Global Fight Against Aids, Tuberculosis, and Malaria, Washington, D.C., United States of America

\* ccollins@theglobalfight.org

**Data Availability Statement:** We uploaded the data files to Dryad. The links are here: - https://datadryad.org/stash/share/

## Abstract

Worldwide, COVID-19 has decimated healthcare systems and highlighted the pressing need to ensure resilience for future pandemics. Given the almost 30% likelihood of another respiratory disease similar to COVID-19 manifesting in the next 10 years, it is imperative to prioritize pandemic preparedness in the immediate future. To this end, tuberculosis (TB) and its management share many similarities to respiratory disease protection, offering an opportunity to dually strengthen TB programs and protect against future pandemics. Looking at data from the World Health Organization (WHO), Global Fund, Our World in Data, and domestic health ministries, it was hypothesized that countries that had better TB program strength going into the pandemic fared better with COVID-19 than those with poorer TB treatment. It was found that countries that recovered their TB program strength (as measured by TB treatment coverage percentages) to or above pre-pandemic levels fared better in terms of COVID-19 pandemic incidence and death. Case studies helped identify common factors across resilient TB platforms in dually successful COVID-19 and TB countries, including community trust, co-epidemic responses that were able to maintain continuity of care, sustained innovation, comprehensive communication across public and private sectors, and maintenance of donor support for TB programs through the pandemic.

## Introduction

Tuberculosis (TB) is the world's top infectious killer, claiming an assumed 1.5 million lives each year [1]. Despite being declared a global health emergency in 1993 [2], a lack of worldwide urgency has resulted in limited progress in combatting TB. An estimated 10.6 million people developed TB worldwide in 2021 [3]; however, despite the alarmingly high burden of TB, in recent years it remained largely overshadowed by global attention focusing primarily on managing the COVID-19 pandemic. As a treatable disease, TB has offered positive

LWiLpGrm7eiTnK45fmYGrqFtRGEv4JB3jb-niXwbdcg - https://doi.org/10.5061/dryad.tht76hf72usp=drive_link.

**Funding:** The authors received no specific funding for this work.

**Competing interests:** The authors have declared that no competing interest exist.

insights and cautionary tales in managing respiratory disease, underscoring the importance of leveraging lessons learned from TB when managing COVID-19 or future respiratory pandemics.

The COVID-19 pandemic highlighted the rapid and widespread impact of respiratory disease, devastating individuals, communities, and economies globally. The pandemic revealed profound weaknesses in global public health systems, highlighting an urgent need for pandemic preparedness and response (PPR). It showcased these threats were realities even in regions unfamiliar with recent pandemic threats. Current models estimate the risk of a COVID-like respiratory virus outbreak in the next decade at nearly thirty percent [4] which could be potentially exacerbated by climate change, increasing international travel, and growing populations.

As respiratory pathogens, TB and COVID-19 share symptom and transmission profiles that allow for common responses and needs for epidemic control. Both TB and COVID primarily attack the lungs and share similar presenting symptoms including cough, fever, shortness of breath, and fatigue. Common risk factors include obesity, preexisting lung conditions, HIV/AIDS, immunosuppression, and cancer and/or chemotherapy exposure [5]. Both diseases are spread through airborne droplets, making masking, social distancing, and other personal protective equipment (PPE) effective measures for limiting transmission [6].

In light of these parallels, TB program infrastructure played an important role in PPR amid the COVID-19 pandemic to enhance disease prevention, detection, and response capabilities for both pathogens in some countries. TB awareness campaigns, surveillance, contact tracing, and community healthcare workers (CHWs) previously utilized for TB response were transitioned to education and prevention of COVID-19 [7]. Bidirectional screening and testing and case notifications allowed healthcare workers to leverage existing TB infrastructure and training to dually detect COVID-19 and TB. Strengthening supply chains and treatment adherence regimens, in addition to expanding previous respiratory pathogen control knowledge, improved treatment accessibility amid the pandemic [3].

Globally, few countries were able to respond to COVID-19 while entirely maintaining their TB program effectiveness [8]. The impacts of the COVID-19 pandemic decimated most TB programs. TB deaths increased in 2020 for the first time in more than a decade. Amidst the pandemic, fewer people were diagnosed, treated, or provided with preventive TB therapies worldwide. TB efforts regressed to 2012 levels, with funding dropping 10% globally [5]

However, few exceptional countries were able to maintain TB program performance through the pandemic. We hypothesized that countries with more resilient and adaptable TB infrastructure were better equipped to respond to COVID-19 throughout all phases of the pandemic. In testing this hypothesis, we found that countries that recovered their TB program strength (as measured by TB treatment coverage percentages) to or above pre-pandemic levels fared better in terms of COVID-19 pandemic incidence and death. Case studies helped identify common factors across resilient TB platforms in dually successful COVID-19 and TB countries, including community trust, co-epidemic responses that were able to maintain continuity of care, sustained innovation, comprehensive communication across public and private sectors, and maintenance of donor support for TB programs through the pandemic.

TB program strength can be a proxy for pandemic preparedness. Investing in TB systems immediately eases the burden of TB while simultaneously fortifying PPR for the future. The following sections of this report discuss statistical and financial findings across COVID-19 and TB resilience, explore identified success factors in positive-recovery countries and corresponding case studies, and provide policy recommendations at the global, national, and local levels to save lives in the next pandemic.

## Methods

This study is a retrospective, observational study considering the correlation between TB program recovery and COVID-19 outcomes across countries. It focuses on 86 countries, selected based on TB burden, eligibility for Global Fund financing, and COVID-19 data availability. COVID-19 data was collected across multiple, robust, open-source, secondary sources, including WHO reports, Our World in Data, national health ministry records, and Global Fund reports. The study relied on publicly available, anonymized data, and thus did not require individual consent, while still adhering to ethical standards for data privacy and integrity. All analysis was completed on Microsoft Excel (v16.78.3) and STATA (v17.0), and the datasets analyzed are included in the supplementary information section. Links to open-source data used are also included at the end.

### I. Identification and classification of countries used in analysis

Eighty-six countries were included in the data calculations. These countries were selected using the Global Fund Eligibility List 2022 [9] and filtered by countries that were simultaneously eligible for GF, TB funding, considered high TB burden, and had available WHO TB data for the 2021 and 2019 TB World Reports [3].

Each country was categorized as 'positive/neutral' or 'negative' recovery for TB using the following methodology:

$$Treatment\ Coverage\ Rate : \frac{\#\ of\ new\ and\ relapsed\ TB\ cases\ notified\ and\ treated}{Estimated\ \#\ of\ incident\ TB\ cases\ in\ the\ same\ year}\ x\ 100$$

$$Delta\ Treatment\ Coverage\ Rate : Treatment\ Coverage\ Rate_{2021} - Treatment\ Coverage\ Rate_{2019}$$

The *Delta Treatment Coverage* variable describes the change in treatment coverage rate between 2021 and 2019. This variable ranged from -52 to +25. Countries with a *Delta Treatment Coverage* value greater than or equal to negative three were characterized as 'positive/ neutral recovery.' Countries with a *Delta Treatment Coverage* value less than negative three were characterized as 'negative recovery.' This was used to assign values to the binary variable *Positive Recovery* as follows:

$$Positive\ Recovery = 0\ if\ Delta\ Tretment\ Coverage \geq -3$$

$$Positive\ Recovery = 1\ if\ Delta\ Treatment\ Coverage \leq -3$$

Countries with the strongest positive recovery included Nigeria, Zambia, Tanzania, and the Democratic Republic of the Congo. Countries with the weakest recovery included Myanmar, Kyrgyzstan, and Kazakhstan (Fig 1).

### II. COVID-19 confirmed cases and deaths

Data on confirmed COVID-19 cases and deaths was collected from Our World in Data, an open-access site under the Creative Commons managed by the Oxford Martin School, Oxford University, and Y Combinator. The goal of this site was to compile and streamline the most accurate COVID-19 statistics and information into one open-source location [10].

For each country, the following information on cases and deaths is provided: new cases confirmed daily, total confirmed cases, change in daily case numbers, deaths reported, change in death rate, and a comparison of death rate to other countries. The data on confirmed cases and deaths comes from the COVID-19 Data Repository by the Center for Systems Science and

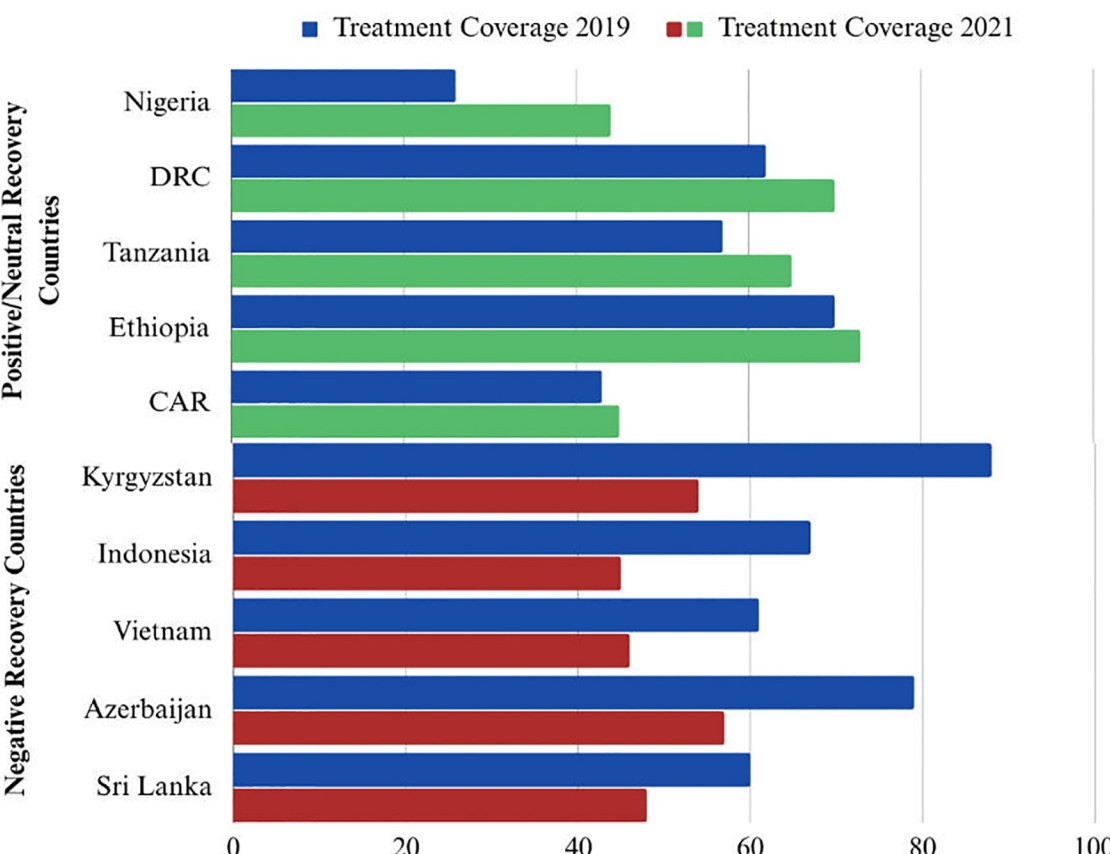

**Fig 1. Countries with the highest and lowest *Delta Treatment Coverage*.** **Total List of Positive/Neutral Recovery Countries**: Algeria, Bangladesh, Burundi, Cameroon, Central African Republic, Chad, Congo, Democratic Republic of Congo, Djibouti, Equatorial Guinea, Eritrea, Ethiopia, Ghana, Guinea, Guinea-Bissau, Iraq, Ivory Coast, Lao PDR, MadagaMali, Mali, Mauritania, Micronesia, Morocco, Mozambique, Niger, Nigeria, Pakistan, Senegal, Sierra Leone, Solomon Islands, Somalia, South Africa, South Sudan, Sudan, Uganda, United Republic of Tanzania, and Zambia. **Total List of Negative Recovery Countries**: Afghanistan, Angola, Armenia, Azerbaijan, Belarus, Benin, Bhutan, Bolivia, Botswana, Cambodia, El Salvador, Eswatini, Fiji, Gabon, Gambia, Georgia, Guyana, Haiti, India, Indonesia, Kazakhstan, Kenya, Kiribati, Kyrgyzstan, Lesotho, Liberia, Libya, Malawi, Malaysia, Marshall Islands, Mongolia, Myanmar, Namibia, Nepal, Nicaragua, Papua New Guinea, Peru, Philippines, Rwanda, Sao Tome and Principe, Sri Lanka, Suriname, Tajikistan, Thailand, Turkmenistan, Ukraine, Uzbekistan and Vietnam.

Engineering (CSSE) at Johns Hopkins University (JHU) and the WHO. The data is accessible on GitHub, and more information on how the WHO and JHU collect their data is accessible on the Our World in Data website. The exact counts used in this report are reflective of reporting in September 2023.

A confirmed COVID-19 case is defined as 'a person with laboratory confirmation of COVID-19 infection' per the WHO [10]. This is *significantly* lower than the suspected number of cases, and thus a drastic underrepresentation of the true toll of COVID-19 [11]. However, using confirmed cases and confirmed deaths helps to standardize data reporting discrepancies across countries in an effort to establish unbiased trends. The same is true of confirmed COVID-19 deaths–this refers to an individual with a confirmed, positive COVID-19 test who then dies directly as a result of infection. Death reporting varies slightly from country to country, as some countries count only in-hospital deaths, while others include deaths at home [6].

### III. Statistical analysis

Welch's two-sample t-test was used to assess differences in means between groups. For each country, data was collected on 23 variables. These included: (1) cumulative confirmed COVID deaths per million, (2) cumulative confirmed COVID cases per million, (3) population in 2021, (4 & 5) TB incidence in 2021 and 2019, (6 & 7) HIV negative mortality in 2021 and 2019, (8 & 9) HIV positive mortality in 2021 and 2019, (10 & 11) TB treatment coverage in 2021 and 2019, (12 & 13) TB case fatality rate in 2021 and 2019, (14 & 15) TB treatment success rate in 2021 and 2019, (16 & 17) total funding in 2021 and 2019, (18 & 19) percentage of domestic funding in 2021 and 2019, (20 & 21) percentage of international funding in 2021 and 2019, (22) percentage of population living in urban slums in 2021, and (23) the Gini Index in 2021.

### IV. Financial data collection, preparation and analysis

The financial analysis of this report included data from the eighty-six countries described in the section above. For the financial analysis, the data was analyzed utilizing Microsoft Excel 16.78.3, before being sorted into funding sources, country expenditures, and allocations or funding gaps as defined by the WHO.

Total international funding percentages were calculated by dividing the total international funding by the total funding overall on a by-country basis. The total funding was calculated by summing the dollar amounts reported from domestic, Global fund, and grant funding sources; the total international funding was calculated by summing only the latter two. To calculate the average Global Fund funding percentages across positive/neutral and negative recovery countries, all Global Fund funding was summed and divided by the 86 number countries included. The absorption rate for positive and negative countries was calculated by dividing their total expenditures by the total funding received. Finally, the dollar spending per TB case incidence in 2021, was separately calculated by first dividing the amount of money for TB programs by case incidence, and then dividing the dollar amount spent on TB drugs by case incidence.

### V. Case studies

Semi-structured interviews were conducted with local representatives with deep knowledge of the TB and COVID co-responses from countries representative of both positive/neutral and negative recovery countries (listed in acknowledgments).

In total, eight interviews were conducted virtually, with representatives from Pakistan, Nigeria, Peru, Liberia, Romania, and Kenya, as well as leaders of TB control organizations including Mercy Corps, TBPPM Learning Network, Centre Universitaire de santé McGill, Partners in Health Liberia, Stop TB Partnership, and the Kenyan Ministry of Health. Each participant was briefed on our intentions following the interviews, and we confirmed no ethics approval was necessary given the non-invasive nature of interview techniques and the use of anonymized data in results. Interviewees/stakeholders were not compensated.

The exact questions asked varied across interviews, depending on time constraints and directions of responses. Generally, questions touched on included:

- How do you think [country's] TB program fared during the COVID-19 pandemic?

- What aspects of [country's] TB program helped to facilitate the COVID-19 response?

- How is [country's] TB program doing presently? Has TB infrastructure recovered from the initial hit of the COVID-19 pandemic?

- What was the donor environment like during the COVID-19 pandemic? Were donors only focused on COVID-19? *(Here it was expressly stated once again that interviewers were independently funded and not working on behalf of the Global Fund).*

- If you were to know another pandemic was coming in five years, what would you want to prepare? What would you prioritize?

- *For leaders of TB control organizations who were not per se country-specific*: What countries fared particularly well during the COVID-19 pandemic? Was this expected or unexpected? What explains the success of these countries? Were these countries particularly good at using existing TB infrastructure to respond to COVID-19?

The information gleaned from interviews with TB experts and on-the-ground contacts was used qualitatively to supplement our discussion and confirm that our quantitative findings reflected the same sentiments as experts in the field, as we feared there might be limitations in data reporting. In addition to these semi-structured interviews, we supported findings with a desk review of available UNDP, WHO, and USAID annual reports and articles regarding TB and COVID-19 progress in addition to funded on-the-ground initiatives. Rather than conducting rigorous coding analysis for data saturation, we focused on qualitatively identifying frequently mentioned terms or themes across interviews. This approach allowed us to integrate themes into our broader analysis.

## Results & discussion

### I. Descriptive statistics

Of the 86 countries examined, there were 33 positive/neutral recovery countries and 53 negative recovery countries (Fig 1). The 86 countries spanned six WHO regions. In Africa, there were 41 total countries, 27 of which were categorized as positive/neutral recovery and 14 negative recovery countries. In the Eastern Mediterranean, there were 8 total countries. Of these, two were positive/neutral recovery and six were negative recovery. In Europe, there were ten countries in total. All ten European countries were negative recovery countries. In Southeast Asia, there were nine countries, with one positive/neutral recovery country and eight negative recovery countries. In the Western Pacific, there were eleven total countries, with three positive/neutral recovery and eight negative recovery countries. In the Americas, there were seven total countries, all negative recovery countries (Table 1).

Countries fell into one of four income categories: lower-income (LI), lower-lower-middle-income (LLMI), upper lower middle-income (ULMI), and upper-middle-income (UMI). There were 21 LI countries, 27 LLMI countries, 15 ULMI countries, and 20 UMI countries.

There was significant variation across all variables. Population ranged from 40,000 on the low end (Marshall Islands) to 1.4 billion on the high end (India). The population across

**Table 1. Breakdown of positive and negative recovery countries by WHO region.**

| Region | Total Country Count | Positive Recovery Count | Negative Recovery Count |
|---|---|---|---|
| All | 86 | 33 | 53 |
| Africa | 41 | 27 | 14 |
| Eastern Mediterranean | 8 | 2 | 6 |
| Europe | 10 | 0 | 10 |
| Southeast Asia | 9 | 1 | 8 |
| Western Pacific | 11 | 3 | 8 |
| Americas | 7 | 0 | 7 |

surveyed countries averaged 46.5 million. The Gini Index, a measure of inequality where zero represents perfect equality and 100 represents perfect inequality, ranged from 24 (Belarus) to 63 (South Africa). Total funding for TB in 2021 ranged from $400,000 USD (Sao Tome and Principe and Equatorial Guinea) to $297 million USD (India). Finally, there was significant variation in pre-pandemic TB program strength, as is shown by the variation in 2019 HIV-negative TB mortality rates and 2019 treatment coverage rate. HIV negative mortality rate (2019) varies from 0.97 per 100,000 (El Salvador) to 98 per 100,000 (Central African Republic). TB treatment coverage in 2019 ranges from 26 (Nigeria) to 97 (Kazakhstan) or 110 (Marshall Islands, though treatment coverage should not feasibly exceed 100).

## II. Countries with stronger TB program resilience had better COVID-19 outcomes

**Positive/neutral TB recovery countries were associated with fewer COVID cases per million.**  The results from Welch's two-sample t-test are summarized in the table below (Table 2). The two populations are positive/neutral recovery countries and negative recovery countries. Using a threshold of *α = 0.05*, there is a significant difference in the cumulative COVID-19 cases confirmed per million between positive/neutral recovery countries and negative recovery countries (Fig 2).

On average, positive/neutral recovery countries had 20,498 confirmed COVID cases per million, while negative recovery countries had 71,164 confirmed COVID cases per million (Fig 2). Thus, positive/neutral recovery countries confirmed an average of 51,000 [95%, CI: (18459, 82872)] fewer COVID cases per capita than negative recovery countries. This represents a nearly fourfold decrease in confirmed COVID cases per capita in countries that were able to recover their TB programs to or above pre-pandemic levels, indicating a lower burden of disease caused by the COVID pandemic in countries that were able to maintain their TB programs from 2019–2021.

**Positive/neutral TB recovery countries were associated with fewer COVID deaths per million.**  The two populations are positive/neutral recovery countries and negative recovery countries. Using a threshold of *α = 0.05*, there is a significant difference in the cumulative COVID-19 deaths confirmed per million between positive/neutral recovery countries and negative recovery countries (Table 3) (Fig 3).

There is a significant difference between cumulative COVID-19 deaths confirmed per million among positive/neutral recovery countries and negative recovery countries. On average, positive/neutral recovery countries had approximately 250 confirmed COVID deaths per million, while negative recovery countries had 850 confirmed COVID deaths per million (Fig 3). Positive/neutral recovery countries confirmed an average of 603 [95%, (CI: 195, 1010)] fewer COVID deaths per million than negative recovery countries. This represents a 3.5-fold decrease in COVID deaths per capita in countries that were able to recover their TB programs to or above pre-pandemic levels, suggesting that the value of TB programs extends beyond TB control itself to pandemic preparedness and response.

**Table 2. Results of Welch's two-sample t-test of positive/neutral and negative recovery countries comparing cumulative COVID-19 cases confirmed per million between both country cohorts.**

| *t* = -3.1372 | *df* = 70.477 | *p-value*: 0.00249 |
|---|---|---|
| *Alt hypothesis*: true difference in means is not equal to zero | | |
| *95% CI*: -82871.94–18459.23 | | |
| *Sample estimates mean of confirmed cases per million in positive/neutral recovery countries*: 20498.80 | | *Sample estimates mean of confirmed cases per million in negative recovery countries*: 71164.38 |

**Fig 2. Average confirmed COVID-19 cases per million for positive/neutral and negative recovery country populations.**
Results of Welch's two-sample t-test.

**Table 3. Results of two-sample t-test between a combined positive+neutral recovery countries cohort and negative recovery countries cohort comparing COVID-19 deaths confirmed per million.**

| $t$ = -2.95 | $df$ = 71.974 | $p$-value: 0.0043 |
|---|---|---|
| *Alt hypothesis*: true difference in means is not equal to zero | | |
| 95% CI: -1010.3803–195.4412 | | |
| *Sample estimates mean of confirmed deaths per million in positive/neutral recovery countries*: 245.67 | | *Sample estimates mean of confirmed deaths per million in negative recovery countries*: 848.59 |

**Regional results confirm the above findings to be true in every surveyed WHO Region.** Six WHO regions were included in the analysis: Africa, the Americas, the Eastern Mediterranean, Europe, Southeast Asia, and the Western Pacific. While there isn't enough power to run t-tests for each region due to smaller sample sizes, in every surveyed WHO region positive/neutral recovery countries averaged fewer confirmed COVID-19 cases and fewer confirmed COVID-19 deaths than negative recovery countries (Figs 4 and 5).

## III. Strong TB financing led to TB program resilience and better COVID-19 outcomes

**Positive/neutral recovery countries received more sustained support from international funding sources, particularly the Global Fund.** Given the strong association between TB program resilience during the pandemic and fewer COVID-19 cases and deaths, we performed further analyses to try and understand the underlying reasons for resilience.

*The importance of balanced financing and Global Fund financing.* Analysis of funding breakdowns by donor source was performed to understand better if donor support for TB programs

**Fig 3. Average confirmed COVID-19 deaths per million for positive/neutral and negative recovery country populations.** Results of Welch's two-sample t-test.

made a difference in strength. Positive/neutral recovery countries received more international funding than negative recovery countries (Fig 6). The Global Fund provided the bulk of TB financing for positive/neutral and negative recovery countries (Fig 7). Positive/neutral recovery countries spent more on TB programs than negative recovery countries and spent a larger share of funding on programs rather than TB drugs per predicted case (Fig 6). This suggests that countries with better-resourced TB programs going into the pandemic had more infrastructure to deploy against both pandemics, leading to more maintainable TB infrastructure and a stronger COVID response. In total, these analyses suggest that more balanced financing between programs and commodities was an indication of greater investment in the underlying health systems critical for COVID-19 control.

*The importance of international financing.* Analysis demonstrates that high-burden TB countries that received more donor support recovered faster than those without. High-income and upper-middle-income countries rely on domestic sources for almost all of their TB investments, while low-to-middle-income countries rely largely on international funding to maintain their TB programs. For positive recovery countries, international funding dipped from 76% to 72% of total TB funding from 2019 to 2021, while negative recovery countries had their international funding share dip from 40% to 24% from 2019 to 2020, before rebounding to 54% of their total funding in 2021 (Fig 6). This correlation between international funding and TB program performance during the acute pandemic period shows that countries that were able to sustain their international funding streams outperformed those that were unable to maintain them.

*Higher proportions of Global Fund financing led to better TB and COVID-19 outcomes.* Specifically to international funding, a large share of Global Fund financial support was associated

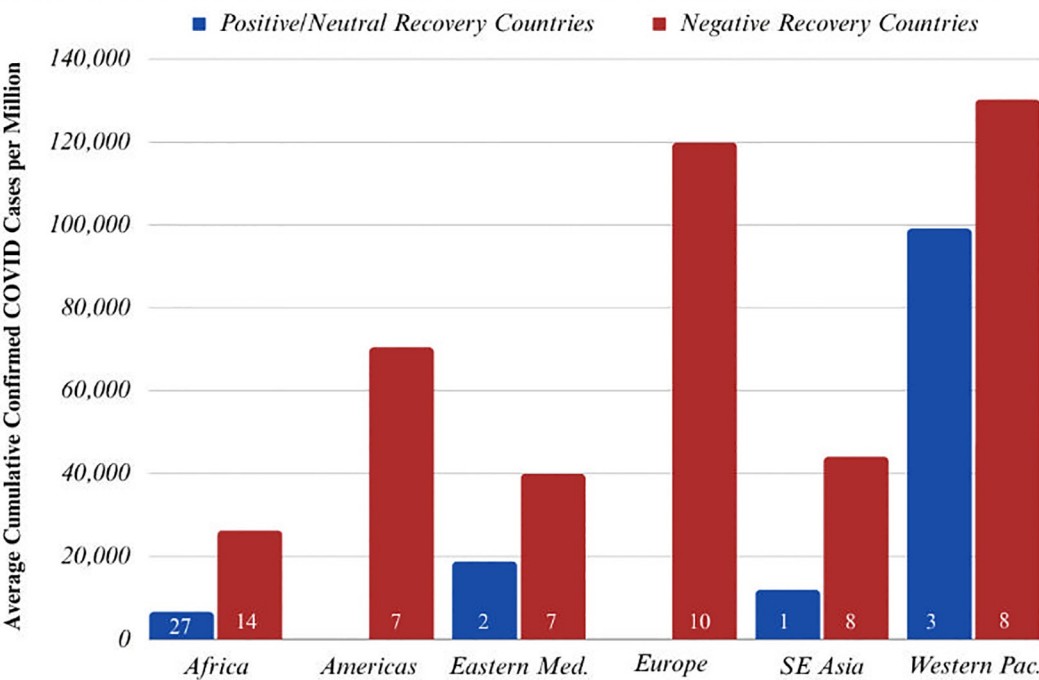

**Fig 4. Average confirmed COVID-19 cases per million for positive/neutral and negative recovery countries, broken down by region.** The number inside the bar indicates how many countries were included in each subgroup.

with a more successful COVID-19 response. For positive/neutral recovery countries, Global Fund funding comprised 64% of total funding in 2019, slightly dipping to 62% by 2020. For negative recovery countries, Global Fund funding constituted 36% and 38% of international funding in 2019 and 2021, with a significant dip in 2020 to 12% (Fig 7). For high-burden TB countries, a positive association existed between a higher proportion of Global Fund funding and TB program recovery during the COVID-19 pandemic.

**Countries that were able to maintain high absorption rates of TB financing were also associated with fewer COVID deaths.** Among countries that received international support, high absorption rates of funding were associated with more successful responses against COVID-19. The absorption rate is the percentage of actual expenditures spent on a nation's TB response compared to the grant budget and was calculated using the countries' expenditure on TB divided by the total grant budget. In 2020, positive recovery countries had an average absorption rate of 91%, compared to 87% for negative recovery countries (Fig 8). Lower absorption rates meant money left on the table or used elsewhere that could be otherwise used for TB prevention, diagnosis, and treatment. High absorption rates not only mean more effective resource allocation but indicate that these countries could benefit from increased financial support.

**Countries with more efficient TB programs were also able to better control COVID.** Positive recovery countries have been able to achieve significantly lower treatment costs per TB case incidence than negative recovery countries. Positive recovery countries spent an average of $0.01 on drugs and $0.05 on programs per case incidence, while negative recovery

### Regional Average COVID Deaths/Mil for Positive/Neutral and Negative Recovery Countries

■ *Positive/Neutral Recovery Countries*   ■ *Negative Recovery Countries*

data shown for regions with one or more countries per recovery category *
number in bar indicates the number of countries included in each calculation *

**Fig 5. Average confirmed COVID-19 deaths per million for positive/neutral and negative recovery countries, broken down by region.** The number inside the bar indicates how many countries were included in each subgroup.

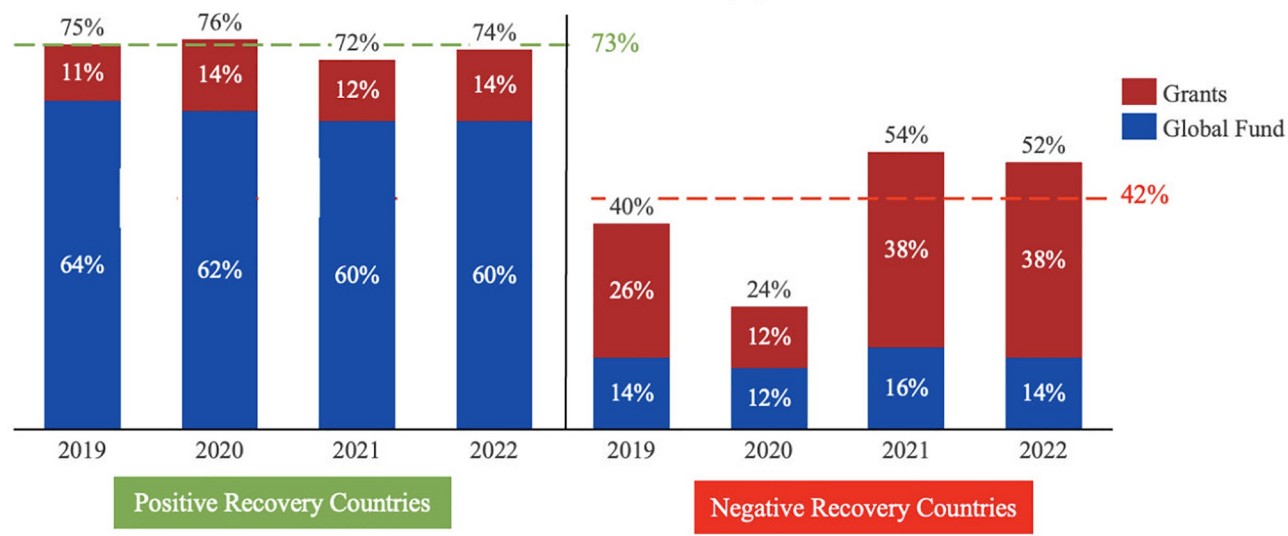

**Fig 6. Total international funding for high-burden TB countries.**

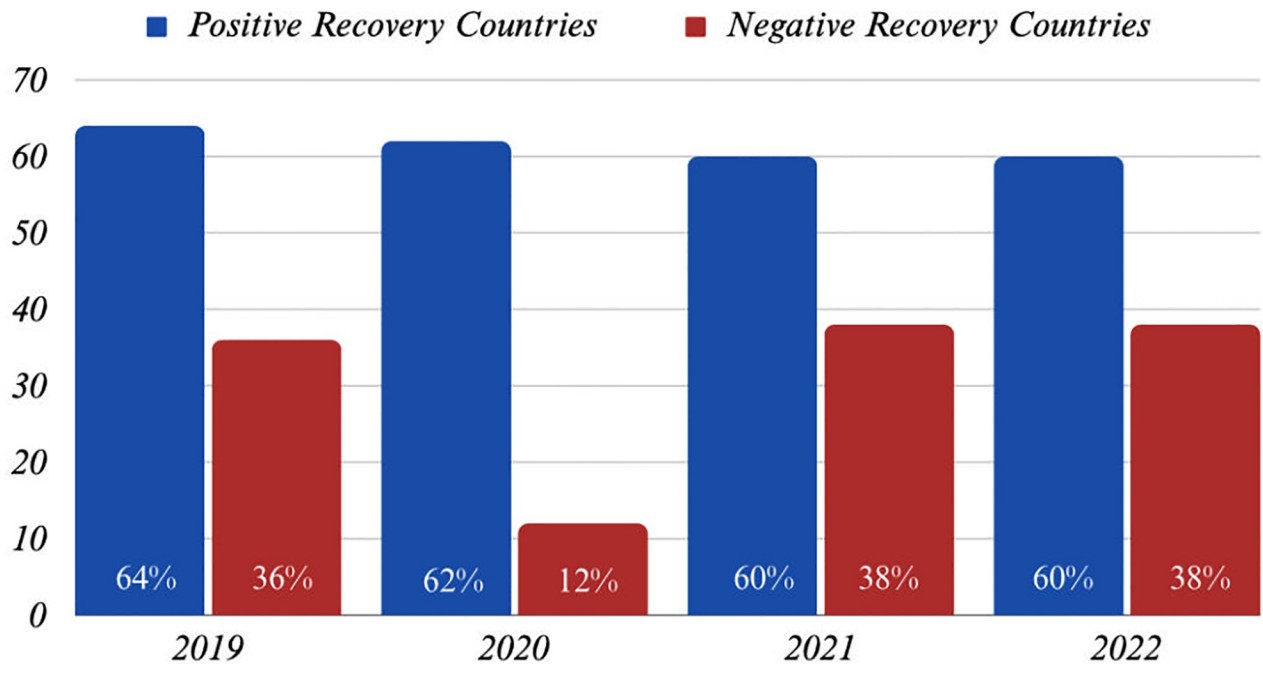

**Fig 7. Global fund funding for high-burden TB countries.**

countries spent nearly double per case incidence: $0.02 on drugs and $0.11 on programs (Fig 9). Potential factors could include the strength of existing healthcare infrastructure and supply chains, efficient utilization of existing resources, and overall pandemic strategies.

## IV. Case study results

From the combination of semi-structured interviews and desk review, five broad themes emerged that appeared to differentiate positive/neutral recovery countries from negative recovery countries. These include:

1. Leveraging existing and local resources and infrastructure within the pandemic response to maintain patient and community trust

2. Prioritizing the creation of a co-epidemic response

3. Maintaining open-mindedness to innovations and adaptations during times of acute pandemic

4. Deepening cross-sectoral coordination in pandemic responses

5. Implementing a flexible, international funding scheme to allow for the adaptation of TB responses to changing local conditions

**Maintaining community trust throughout the pandemic response.** Previous studies have validated the importance of trust more than almost any other factor in how well a country

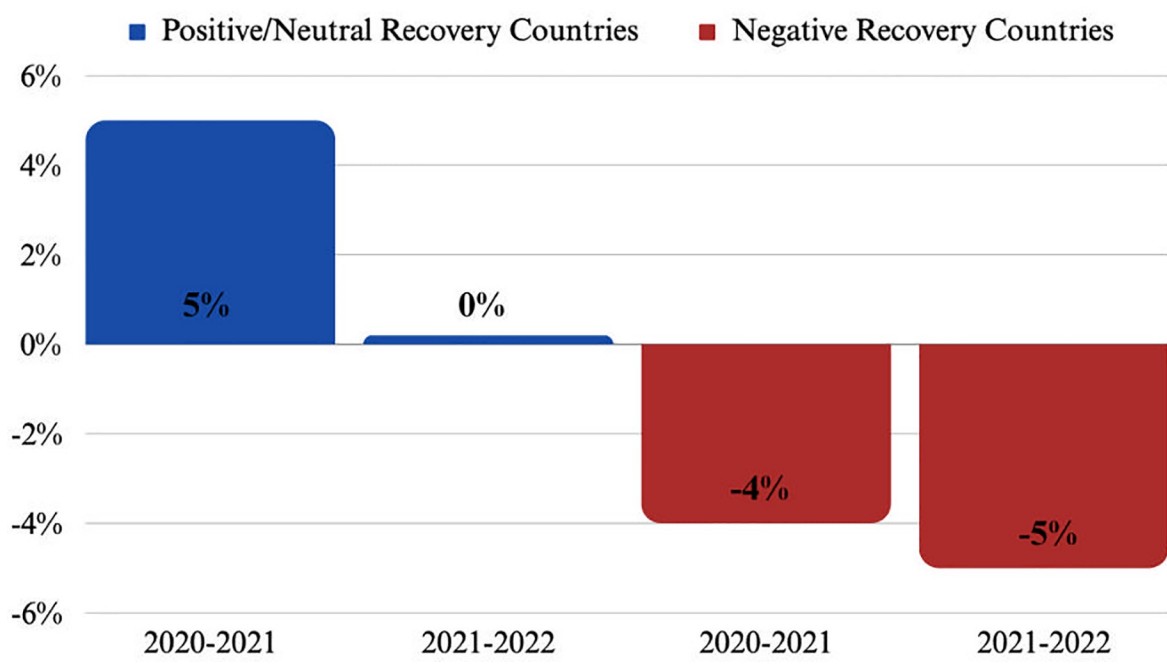

**Fig 8. Absorption rates for high-burden TB countries.**

was able to control COVID, particularly in the earliest phases of the pandemic [12]. Positive recovery countries prioritized using local community groups and already established trusted relationships to ensure continuity of care for existing TB care and navigate around disruptions caused by lockdowns. In Sierra Leone, public health authorities and international agencies, such as Medecins Sans Frontieres (MSF), piloted programs to train locally trusted traditional

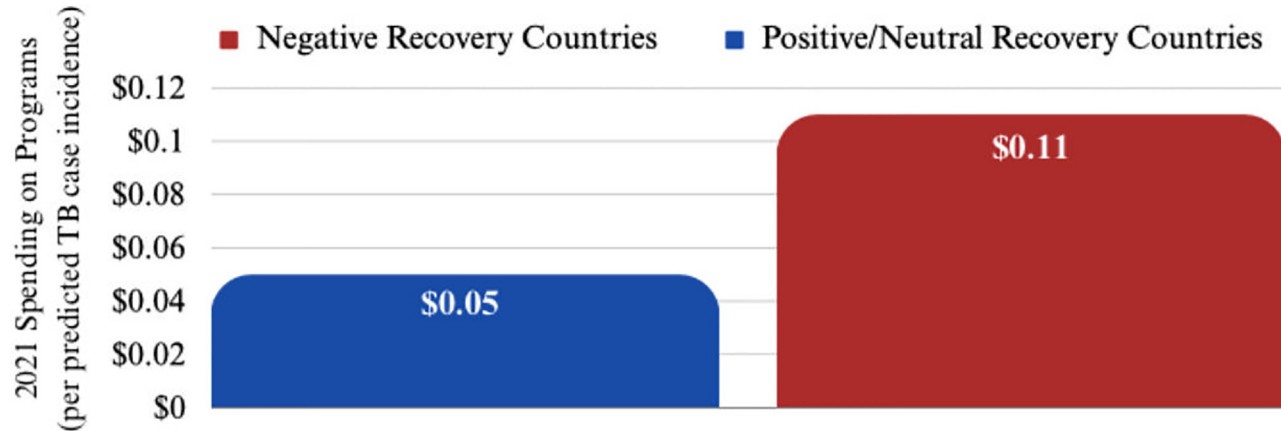

**Fig 9. Spending categories distribution for positive and negative recovery countries.**

healers and workers in detecting potential TB symptoms in patients and referring them to local clinics. Beyond increasing the TB detection rate, this built locals' trust in public health efforts due to the involvement of authority figures already trusted by the community. In Pakistan, understanding that private health clinics were generally more trusted by the populace due to their embedded nature in local communities, international NGOs created and funded local initiatives to ensure continuity of care for their TB patients even during nationwide private health care facility shutdowns [12].

**Creating co-epidemic pandemic responses.**   During the acute phase of the COVID crisis, many countries diverted funding and staff away from TB and other basic health programs towards COVID-19 management. Resilient countries, however, took the approach of treating both TB and COVID-19 as co-epidemics and continued to support and resource TB and COVID-19 programs together, avoiding a disease-siloed response.

As an example, bidirectional screening played a large part in the maintenance of Nigeria's TB response during COVID-19. Wellness on Wheels Mobile Units were equipped with GeneXpert instruments and digital X-ray machines that allowed for mobile case finding, referring TB-positive individuals to treatment health facilities while dually screening for COVID-19. They played a key role in increasing TB case notification rates in Nigeria throughout the pandemic, optimizing patients' time and healthcare resource utilization while expanding patients being tested for TB during the acute pandemic period [13].

**Deepening coordination in building effective pandemic responses.**   Most countries took an all-of-society approach to COVID-19, incorporating various levels of government and private partners into the strategy and management of the pandemic response; the success of these strategies suggests this approach should be replicated against other diseases. As observed in Nigeria, increased public-private coordination resulted in the alignment of COVID-19-era TB patient care procedures, ensuring continuity of care for all patients despite pandemic restrictions. In the Central African Republic (CAR), the WHO coordinated the development of local micro-plans to effectively combat TB. These included targets and reporting gaps; mapping and inventory exercises; and specimen transport strategies. Additionally, twenty-three GeneXpert machines were purchased and distributed amongst regional facilities in the CAR to facilitate increased access to high-quality laboratory COVID-19, TB, and HIV testing. This decentralizing pandemic response strategy led to a four-fold increase in laboratory TB tests conducted, with an increase anticipated as COVID-19 testing capacity requirements decrease.

**Encouraging innovation and technological adaptation in times of acute pandemic.** Several countries rapidly engaged in innovative and adaptive solutions that bypassed limitations imposed by the pandemic. Such solutions were encompassed in the integration of local transportation networks and infrastructure into the national TB response, as well as swift implementation of eHealth systems for expanded access to health diagnostics, increased public investment and accountability, and enhanced care.

The case of Pakistan is notable due to the engagement of the traditional TB response with delivery and courier systems already in place. The National TB Control Programme meticulously located all registered patients and utilized courier services like the Pakistan Postal Service, and private delivery services such as Uber, Bykia, or other local arrangements to ensure uninterrupted drug supply [14]. The program also revolutionized sputum transport protocols through digitized transportation systems that operated at the district level and included an online tracking dashboard [14]. Nigeria also exemplified rapid adaptation throughout the pandemic with its robust eHealth technologies. The mobile app *TB Screening and Tracking for Accelerated Referral and Reporting* enabled providers to report TB service delivery data and monitor performance in real time [15]. Additionally, it fast-tracked the distribution of additional anti-TB drug stocks to private facilities.

*Ensuring Resilient, Flexible International Financing Streams for TB Programs.* Fundamental to the achievements of the aforementioned case studies are the sustained funding sources supplied by international financing and partnership organizations, specifically the Global Fund. Nations that have received investments from the Global Fund have historically demonstrated success in improving TB-related statistics. In India, an investment of approximately $1.1 billion yielded a 22% decrease in TB incidence rate per 100,000, an 8% decrease in TB deaths, and a 38% increase in HIV and TB patients on antiretroviral treatment. In Nigeria, an investment of approximately $450 million yielded a 17% increase in MDR-TB treatment success rate, while also increasing HIV and TB patients on antiretroviral treatment by 58% [16]. Following COVID-19, supplemental investments by the Global Fund from 2021–2023 through the form of grants, totaling approximately $154 million, have further contributed to Nigerian access to comprehensive, high-quality, patient-centered, community-owned TB services. To date, the Global Fund has invested nearly $10 billion in addressing TB since 2002. Considering that the Global Fund provides as much as 76% of all international funding towards TB treatment, the tremendous significance of ensuring sustainable funding sources for the Global Fund cannot be understated [16].

## Strengths and limitations

The most significant limitation of this study is the lack of uniform, high-quality COVID data. Despite using the most credible sources obtainable, there exists a risk of inconsistency in how COVID case and death reporting differed across countries. Prior infrastructure may have additionally impacted countries' ability to test, detect, and report COVID cases in the early days of the pandemic.

We have attempted to compensate for this by triangulating across multiple datasets of estimates of COVID incidence and mortality and including a broad analysis of 86 countries. We have also used the most extensive data range available in an attempt to eliminate biases across timelines. We assume that from these efforts any major data weaknesses will be averaged out. As for the lack of uniformity, we noted a difference in data reporting for COVID-related deaths, as some countries considered death from a pre-existing factor (e.g. HIV, cancer) that was exacerbated by current COVID infection as a COVID-related death, and others did not. To mitigate biases introduced by this, in addition to COVID deaths, we considered COVID cases and overarching estimates of excess mortality during the COVID pandemic and found aligning results.

Pandemics are continuously evolving, and the dynamics of global health are influenced by a complex, interconnected web. This poses the risk of confounding variables that could influence healthcare outcomes, such as political stability, climate, or government response to COVID. In an effort to control for as many of these factors as possible, we segmented our data by region, by income, and by political regime, and found the same trends held (not all t-test results are shown for succinctness).

The main strengths of our study are the rigorous statistical analyses, the incorporation of robust data sets, and the large scope of countries surveyed. Our findings held across the 86 countries investigated, despite variations in data availability. The integration of qualitative and quantitative data in our study also served as a major strength, as we were able to validate our quantitative results by hearing on-the-ground reports of the same trends.

## Conclusion and policy recommendations

There is a pressing need to address pandemic preparedness given the likelihood of another respiratory pandemic in the near future. Recent experiences with COVID-19 have highlighted

TB platforms as a mechanism that can be rapidly pivoted to detect and respond to emerging respiratory pandemics, making leveraging TB programs an imperative component of pandemic preparedness plans [6]. COVID-19 has shown that respiratory pandemic control requires the same pillars as responses to TB, namely surveillance capability, detection/testing, case notifications, and treatment. These commonalities show interoperability between well-established TB programs and novel pandemic management [17].

This analysis highlights that high-burden TB countries with strong case-detection capabilities going into the COVID pandemic had fewer COVID-related deaths and cases relative to their less capable peers (Figs 1 and 2). Additionally, high-burden TB countries that were able to restore case-detection capabilities in 2020 and 2021 (during the pandemic) to levels equal to or above 2019 (pre-pandemic) also experienced fewer COVID-related deaths and cases relative to regional peers (Figs 4, 5 and 9). These trends were statistically significant across a diverse set of countries, suggesting that countries that were able to manage their TB programs effectively during the acute phase of the pandemic were also able to extend these capabilities to manage COVID-19 as well (Figs 1–9).

To achieve these outcomes, potential success factors included: leveraging existing and local resources and infrastructure within the pandemic response to maintain patient and community trust; prioritizing the creation of a co-epidemic response; maintaining open-mindedness to innovations and adaptations during times of acute pandemic; deepening cross-sectoral coordination in pandemic responses; and implementing a flexible, international funding scheme to allow for the adaptation of TB responses to changing local conditions. This is with a particular focus on community-level activities that leverage trust-based systems already established in high-burden areas, all while maintaining an innovative mindset that allows for adaptation, especially in times of acute pandemic.

These recommendations vary depending on their intended audience. Domestic governments should encourage other donor countries to invest in TB programs with immediate and long-term use, especially community-level initiatives and trust-based systems. These include investing in health systems, community health workers, surveillance, case detection and notification, and treatment success. Health ministries should focus on dual-investment programs for TB and respiratory pandemic preparedness, while also bolstering community-level and trust-based initiatives already respected within communities. International donors need to treat TB and future respiratory pandemics as a co-epidemic, continuing to reinforce investment in things that can be used for both purposes and support countries in their pandemic preparedness. As was shown, it is evident that coordination with international funding is imperative to maintaining sustainable tuberculosis programs that can be leveraged for pandemic preparedness. For members such as the Global Fund, this includes ensuring funding goes directly towards TB programs, and that funding is replenished efficiently when allocated to non-TB programs. Community advocates should continue to hold donors accountable, maintaining focus on TB and future pandemic preparedness.

In all, these recommendations all have the imperative undertones of maintaining an innovation mindset, co-epidemic approach, and funding of programs emphasizing the dual benefits of TB systems. TB killed 1.6 million people alone in 2021 and continues to do so yearly, with the number only growing. Investing in TB programs offers the chance of a simultaneous victory both for TB control and future respiratory pandemic preparedness.

## Acknowledgments

This analysis was partially submitted in fulfillment of the capstone project requirements for a bachelor's degree in Global Affairs at the Jackson School of Global Affairs at Yale University.

We would like to thank Autumn Eastman (Partners in Health Liberia), Aamna Rashid (Mercy Corps), Enos Masini (Stop TB Partnership), Azhee Akinrin (TBPPM Learning Network), Luz Villa Castillo (McGill University Research), and Eliud Wandwalo.

## Open-source data websites

- https://ourworldindata.org/covid-deaths#deaths-from-covid-19-background

- https://github.com/owid/covid-19-data

- https://app.powerbi.com/view?r=eyJrIjoiMGIwZDUzMmItODE5Yi00YjAzLTliMGEtNGVhMGVlYzA4YWVkIiwidCI6ImY2MTBjMGI3LWJkMjQtNGIzOS04MTBiLTNkYzI4MGFmYjU5MCIsImMiOjh9

- https://www.who.int/publications/i/item/9789240037021

- https://ourworldindata.org/coronavirus

- https://www.who.int/teams/global-tuberculosis-programme/tb-reports/global-tuberculosis-report-2022

- https://github.com/owid/covid-19-data

- https://reliefweb.int/report/world/global-tuberculosis-report-2019#:~:text=WHO%27s%20latest%20Global%20TB%20Report%2C%20released%20today%2C%20highlights%20that%20the,getting%20the%20care%20they%20need

## Author Contributions

**Conceptualization:** Whitney Bowen, Ho Tri, Sebastian Romero, Roaa Shaheen, Victoria Kipngetich, Nick McGowan, Sungho Moon, Esha Bhattacharya, Robert Hecht, Shan Soe-Lin, Chris Collins.

**Data curation:** Whitney Bowen.

**Formal analysis:** Whitney Bowen, Ho Tri.

**Investigation:** Whitney Bowen.

**Methodology:** Whitney Bowen.

**Supervision:** Shan Soe-Lin, Chris Collins.

**Visualization:** Whitney Bowen.

**Writing – original draft:** Whitney Bowen, Ho Tri, Sebastian Romero, Roaa Shaheen, Victoria Kipngetich, Nick McGowan, Sungho Moon, Esha Bhattacharya.

**Writing – review & editing:** Whitney Bowen, Ho Tri, Sebastian Romero, Roaa Shaheen, Victoria Kipngetich, Nick McGowan, Sungho Moon, Esha Bhattacharya, Robert Hecht, Shan Soe-Lin, Chris Collins.

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
