## [Decision Letter · Decision Letter 0]

22 Mar 2024

PGPH-D-24-00269

Leveraging Tuberculosis Programs for Future Pandemic Preparedness: A Retrospective Look on COVID-19

Dear Dr. Collins,

Thank you for submitting your manuscript to PLOS Global Public Health. After careful consideration, we feel that it has merit but does not fully meet PLOS Global Public Health’s publication criteria as it currently stands. Therefore, we invite you to submit a revised version of the manuscript that addresses the points raised during the review process.

Please carefully review and address the feedback from both peer reviewers, particularly Reviewer 2's comments regarding the methods and presentation of the data. Please submit your revised manuscript by May 06 2024 11:59PM. If you will need more time than this to complete your revisions, please reply to this message or contact the journal office at globalpubhealth@plos.org. Please include the following items when submitting your revised manuscript:

We look forward to receiving your revised manuscript.

Kind regards,

Sanjana J. Ravi, PhD, MPH

Academic Editor

Journal Requirements:

Additional Editor Comments (if provided):

Reviewers' comments:

Reviewer's Responses to Questions

**Comments to the Author**

1. Does this manuscript meet PLOS Global Public Health’s publication criteria? Is the manuscript technically sound, and do the data support the conclusions? The manuscript must describe methodologically and ethically rigorous research with conclusions that are appropriately drawn based on the data presented.

Reviewer #1: Partly

Reviewer #2: Yes

2. Has the statistical analysis been performed appropriately and rigorously?

Reviewer #1: I don't know

Reviewer #2: No

3. Have the authors made all data underlying the findings in their manuscript fully available (please refer to the Data Availability Statement at the start of the manuscript PDF file)?

Reviewer #1: No

Reviewer #2: Yes

4. Is the manuscript presented in an intelligible fashion and written in standard English?

Reviewer #1: Yes

Reviewer #2: Yes

5. Review Comments to the Author

Reviewer #1: Thank you for taking the time to prepare and submit the manuscript. I have some comments, as indicated below, for the authors to consider.

It would be good to reorganise the text in the introduction section to include the context of TB before COVID-19. This would help readers to recognise that the focus of this study is about TB control efforts as influenced by the COVID-19 pandemic.

In section I of the Methods section, some equations are not displayed in the downloaded PDF file such as the part about the categories of ‘positive/neutral’ or ‘negative’ recovery for TB.

In section V of the Methods section on case studies, please provide more details about the process of conducting the semi-structured interviews such as sample size and participant recruitment process. Please describe how these results from the case studies would be used alongside the other methods that are focused on quantitative work. Please state the details of the ethics approval. Please include relevant methodological details such as explaining the coding process and saturation. Also, suggest including a summary of topics or questions asked. It would be good to include details about the process of analysing the data.

Please include relevant quotes from the interviewees in the results section. It is unclear to me which parts of the sentences are attributed to desk review or semi-structure interviews. These are considered qualitative data that should be included to offer a comprehensive analysis.

Please improve the discussion regarding the strengths and limitations of this study.

Overall, I encourage the authors to reflect on why these findings matter in epidemic and pandemic research.

Thank you. I enjoyed reading the article.

Reviewer #2: Abstract - The sentence “Looking at data from the..” appears incomplete. Should the period be replaced with comma?

Methods – The Data collection and sources of data are very well explained but please specify the study design in the start of the section. Provide more details.

Line 148 to 151 – Is there a text or figure that needs to appear here?

Line 153 – Not sure, if the line 148 was meant to cover this. But please provide context for Delta treatment coverage. Why are the countries categorized using this variable? Is there a reference for this variable of choice?

Figure 1 – This does not correlate with the text/List in Lines 160-162. Why only few countries are chosen in this chart?

Line 232 - Section V. Case Studies – Is this the whole study design? Or is it the last phase after the review of data?

Line 232 – How were the interviews conducted? In person or virtual? Were the stakeholders compensated? How was this data analysed? Were they coded? Did you use a software for analysis?

Line 237 – Variety of TB control organizations? With Organizations names (MC, TBPPM, PIH, etc.). Do you mean these organizations (MC, TBPPM, PIH, etc.). were the “variety” that you interviewed? Or did you partner with these organizations (MC, TBPPM, PIH, etc.) and interview other organizations?

Line 232 to 239 – Why is the case study different from the data collection mentioned above? Why were they all not combined together to report the results?

Line 245 – Surveyed? Or Reviewed?

Line 245 to 253 – These numbers seem interesting. Given that this “Data treatment coverage” categorization is important for your study, it will be good to represent this data and the variation among contents in a data matrix.

Line 346 and Line 372 – How are they different from each other? Both talk about the importance of Global Fund. What is the reason for 2 categorizations?

Line 398 – The treatment costs mentioned here, does it denote COVID treatment of TB treatment?

Line 414 to 420 – What do these themes mean? Provide more description.

Line 414 to 420 – Consider combining these themes with the themes you have identified from the data review to form meta themes and report the results section?

Line 422 – Is this the discussion section of the paper?

Line 440 – The reference, Padmapriyadarshini et al does not seem to be there in the references list (Line 568)

References – Please verify all references. Some of the references mentioned in the text are not on this list. Few references have the PDF names (e.g. Line 581,582,588,etc.) without any further details. Which organization? Available online? How was it accessed? Etc. Please review citation guidelines.

6. PLOS authors have the option to publish the peer review history of their article (what does this mean?). If published, this will include your full peer review and any attached files.

**Do you want your identity to be public for this peer review?** For information about this choice, including consent withdrawal, please see our Privacy Policy.

Reviewer #1: No

Reviewer #2: No

---

## [Editor Report · Decision Letter 1]

5 Jul 2024

Leveraging Tuberculosis Programs for Future Pandemic Preparedness: A Retrospective Look on COVID-19

PGPH-D-24-00269R1

Dear Mr. Collins,

We are pleased to inform you that your manuscript 'Leveraging Tuberculosis Programs for Future Pandemic Preparedness: A Retrospective Look on COVID-19' has been provisionally accepted for publication in PLOS Global Public Health.

Best regards,

Sanjana J. Ravi, PhD, MPH

Academic Editor